# Consumption of Biscuits with a Beverage of Mulberry or Barley Leaves in the Afternoon Prevents Dinner-Induced High, but Not Low, Increases in Blood Glucose among Young Adults

**DOI:** 10.3390/nu12061580

**Published:** 2020-05-28

**Authors:** Mai Kuwahara, Hyeon-Ki Kim, Mamiho Ozaki, Takuya Nanba, Hanako Chijiki, Mayuko Fukazawa, Jin Okubo, Yui Mineshita, Masaki Takahashi, Shigenobu Shibata

**Affiliations:** 1Laboratory of Physiology and Pharmacology, School of Advanced Science and Engineering, Waseda University, Shinjuku-ku, Tokyo 162-8480, Japan; kmykmya@akane.waseda.jp (M.K.); hk.kim@aoni.waseda.jp (H.-K.K.); m-yk_1426@fuji.waseda.jp (Y.M.); 2Graduate School of Advanced Science and Engineering, Waseda University, 2-2 Wakamatsu-cho Shinjuku, Tokyo 162-8480, Japan; mo_u2@ruri.waseda.jp (M.O.); n-x.t.x-n@asagi.waseda.jp (T.N.); hnk-1022@akane.waseda.jp (H.C.); yu8m1omo5iwo6@suou.waseda.jp (M.F.); zin_eile_0kb@akane.waseda.jp (J.O.); 3Institute for Liberal Arts, Tokyo Institute of Technology, 2-12-1 Ookayama Meguro-ku, Tokyo 152-8550, Japan; takahashi.m.bp@m.titech.ac.jp

**Keywords:** snack, dietary fiber, postprandial glucose, young adults

## Abstract

We examined the impact of consuming biscuits with a beverage of powdered mulberry or barley leaves in the afternoon on postprandial glucose levels at dinnertime among young adults. A total of 18 young adults participated in a partially double-blinded, randomized crossover trial over 2 weeks, consuming either: (1) no biscuits; (2) a biscuit; (3) a biscuit with a beverage of powdered mulberry leaves; or (4) a biscuit with a beverage of powdered barley leaves, as an afternoon snack followed by a standardized test dinner. Glucose levels were recorded after each meal. Results showed intake of biscuits with a beverage of mulberry and barley leaves significantly reduced postprandial rises in glucose after their immediate consumption and dinner, though there was no direct relationship between the glucose levels at the two meals. Compared to those with low glucose levels, participants with high glucose levels at dinner showed a stronger second meal effect, that was attributed to the mulberry or barley leaves, and were also more likely to have lean body weights and prefer evenings. Our findings indicate that eating snacks alongside mulberry or barley leaves is an effective way to suppress postprandial glucose levels in young adults with high glucose levels who prefer evenings.

## 1. Introduction

Most people consume three meals per day: breakfast, lunch and dinner. Foregoing breakfast has been considered a risk factor for obesity and type 2 diabetes because it often leads to overeating and increased insulin resistance [1,2], whereas eating breakfast with adequate amounts of protein and fiber helps to keep blood sugar levels low throughout the day [3,4,5]. The “second meal effect” maintains that the previous meal affects the postprandial glucose levels incurred by the following meal, and in a previous investigation, it was found that eating high-fiber snacks in the late afternoon prevented rises in blood sugar at dinner [6]. Interestingly, hyperglycemia tends to be most commonly observed at or after dinner, because of the body’s prolonged release of insulin [7,8,9,10], so efforts to further improve upon the nutrient quality of foods consumed before this meal are of great interest.

In Japan and China, powdered mulberry and barley leaves are functional ingredients widely used to prevent increases in glucose levels [11,12], being rich in polysaccharides and dietary fiber [13]. Mulberry leaves, in particular, contain deoxynojirimycin, which inhibits the breakdown of starch and other disaccharides by alpha-glucosidase into glucose. Studies have found this compound is effective at lowering the glycemic indexes of ingested carbohydrates, for healthy volunteers [14] and patients with type 2 diabetes [15] alike. Additionally, both types of leaves possess unique viscous dietary fibers that become thick when mixed with liquids, resulting in a reduced rate of glucose digestion and absorption [16,17].

To our knowledge, there has been little research as to whether supplementing snack foods with mulberry and barley leaves improves postprandial glucose levels. Moreover, it is unknown whether attenuation of blood glucose by a healthy snack correlates with its attenuation at dinner, and if the second meal effect occurs more frequently in individuals with greater increases of blood glucose compared to those with moderate increases. In the current study, a partially blinded crossover trial was conducted, wherein participants were given biscuits and water, water with powdered mulberry leaves, or water with powdered barley leaves to consume before a test dinner. Thus, we hope biscuits can become a healthy snack if taken in combination with a healthy beverage. Continuous glucose monitoring (CGM) occurred at and after each snack and dinner. We hypothesized that there is a direct relationship between glucose levels at snack time and those at dinner time, and also there is a direct relationship between high glucose levels and the second meal effect.

## 2. Materials and Methods

### 2.1. Participants

A total of 18 university students (ages 23.1 ± 0.4 years, 11 males and 7 females) were enrolled. Prior to beginning the study, individuals completed a questionnaire on physical activity, diet, lifestyle habits and overall health. All were generally healthy and took no medications. The protocol was approved by the Ethics Committee of Waseda University (2018-074) and conducted according to the guidelines delineated by the Declaration of Helsinki. Human trial of the present study is registered at UMIN Clinicalinic Trials registry (UMIN-CTR) as UMIN000033480 [18]. None of the study participants were trained athletes competing in any sporting event, but some were recreationally active. Three participants were excluded from the diet survey analysis because of wrong record. Consequently, 18 participants were included in all analyses, excluding the diet survey analysis (*N* = 15).

### 2.2. Test Sessions

The total duration of the experimental period was two weeks. A partially blinded (mulberry or barley) randomized crossover design was used. Each participant attended four sessions in a randomized order, receiving either: (1) no biscuit with water (control group); (2) a biscuit with water (vehicle group); (3) a biscuit with the mulberry drink (blinded) (mulberry group); or (4) a biscuit with the barley drink (blinded) (barley group) (Figure 1A). Each session was conducted with at least a two-day washout period during the two weeks. For all sessions, participants were instructed to consume their snack during the late afternoon (4 h after lunch and 4 h before dinner). The mealtime for each session is shown in Table 1. During the test sessions, a test meal was served.

### 2.3. Anthropometric Measurements

Following each session, all participants were required to visit the laboratory following a 12-h overnight fast for anthropometric measurements baseline, and again two weeks after the completion of the study. Height was measured to the nearest 0.1 cm using a wall-mounted stadiometer (Seca 213, As One Corporation, Japan). Body mass was measured to the nearest 0.1 kg using a digital balance (InBody 270, InBody Co. Ltd., Tokyo, Japan). Muscle volume, fat mass and body fat percentage were measured by direct segmental multifrequency (20 kHz to 100 kHz) bioelectrical impedance (Inbody 270, Inbody Inc., Tokyo, Japan).

### 2.4. Dietary Intake Assessments

A Food Frequency Questionnaire (FFQ) assessed nutrient intakes. FFQs are widely used across the globe [19], and ours consisted of 29 foods with questions pertaining to their frequency of consumption and when they were eaten. Furthermore, we recorded the intake frequency at breakfast, lunch, and dinner, calculating the energy intake for each meal [20]. Note: participants were asked not to change their dietary patterns throughout the testing period. Unfortunately, three participants were excluded from the data, because they did not record it.

### 2.5. Blood Glucose Evaluations

A CGM System (FreeStyle Libre Pro Flash Glucose Monitoring System) was worn, and glucose levels were monitored every 15 min. For all sessions, glucose levels and area under the curve (AUC) were calculated 3 h after snack and dinner intake. AUC (mg/dL·h) was calculated from 0 to 180 min. Mean glucose level (mg/dL) throughout experiment was also calculated on a daily basis. Additionally, from the median AUC value of the postprandial blood glucose level at dinner in the control session, we examined blood glucose level changes in each session, and divided them into high-(*N* = 9) and low-(*N* = 9) glucose groups.

### 2.6. Physical Activity Assessments

A triaxial accelerometer (Active style Pro HJA-750C, Omron Co., Ltd., Kyoto, Japan) was worn. Participants were instructed to wear the accelerometer at all times except when showering. Data was only considered valid if a participant wore the accelerometer for at least 10 h (600 min) per day, on at least two weekdays and one weekend day. Note: participants were asked to refrain from remaining inactive or participating in high-intensity physical activity throughout the testing period.

### 2.7. Circadian Rhythms

Circadian rhythms were evaluated using the Horne–Ostberg Morningness–Eveningness Questionnaire (MEQ) [21]. We used the MEQ score to determine chronotype. This consists of 19 questions related to preferred sleep time and daily performance. In the current experiment, score range was from 32 to 56, and high score and low score means preference for the mornings (morningness) and preference for the evenings (eveningness), respectively.

### 2.8. Nutritional Information for Snacks and Dinner Meals

Powdered mulberry and barley leaves were purchased from Toyotama Healthy Food Co. (Tokyo, Japan). Per their nutritional labels, the mulberry leaves were found to contain deoxynojirimycin (0.21%), galactoside form of deoxynojirimycin (0.10%) and fagomine (0.03%). None occurred in the barley leaves. Their basic nutrition facts are shown in Table 2.

Biscuits were purchased from Seikatsu Shikou Co. (Ibaraki, Japan). Their basic nutrition facts are shown in Table 3. On a trial day, participants consumed a biscuit (weight: 4 g/sheet, shape: a circle with a radius of 2.5 cm) with a drink consisting of 3 g of mulberry or barley powder dissolved in water, followed by test dinner. Test dinner meal was made by combination of foodstuff from Otsuka Seiyaku Co. (Tokyo, Japan; energy: 400 kcal, protein: 8.7 g, fat: 22.4 g, carbohydrate: 41.7 g), Sato Shokuhin-kogyo Co. (Tokyo, Japan, energy: 147 kcal, protein: 2.5 g, fat: 0.1 g, carbohydrate: 33.9 g), Morinaga Seika Co. (Tokyo, Japan; energy: 194 kcal, protein: 10.5 g, fat: 10.5 g, carbohydrate: 14.8 g), and Marumiya Shokuhin-kogyo Co. (Tokyo, Japan; energy: 11 kcal, protein: 0.6 g, fat: 0.6 g, carbohydrate: 1.0 g). A survey of typical dinner diet administered through the Food Frequency Questionnaire (FFQ) in 15 participants demonstrated that mean nutrient compositions were energy (640 Kcal), protein (29 g, 19% based on energy), fat (30 g, 44%) and carbohydrate (56 g/Dietary fiber: 4.6 g, 36%,). A previous study indicated that high percentage of carbohydrate loading can potentially increase postprandial glucose in healthy adults [22,23]. However, participants in the current experiment showed low ratios of carbohydrate (36%), and we decided to serve standard dinner (Table 3) with a high ratio of carbohydrate (50%).

Finally, participant characteristics in the current experiment are summarized and presented in Table 4.

### 2.9. Statistical Analyses

All parameters were tested for normality using the Kolmogorov–Smirnov test. To compare differences in postprandial glucose levels after snack intake between the sessions, a repeated-measures one-way analysis of variance with a Tukey’s test was used. In order to determine relationships between physiological factors, glucose levels and AUC, a Pearson’s correlation coefficient was calculated. To investigate the physical characteristics between high- and low-glucose groups, unpaired t-tests were performed (after confirmation of data normality). All data were analyzed using Predictive Analytics Software for Windows (SPSS Japan Inc. Tokyo, Japan); *p*-value < 0.05 indicated statistical significance.

## 3. Results

### 3.1. Intake of Biscuits Increased Postprandial Blood Glucose Levels

Consumption of the snack biscuit 4 h after lunch increased glucose levels, though levels returned to basal level within 3 h (Figure 1A,B). Eating a biscuit (vehicle group) as a snack significantly increased AUC compared with not eating a biscuit (none group) (Figure 1C). Intake of mulberry or barley leaves alongside the biscuit significantly attenuated rises in glucose (Figure 1C). At dinner time, glucose levels by AUC were higher in the none and vehicle groups, compared to the mulberry and barley groups (Figure 1D).

### 3.2. Correlations between Glucose AUCs Revealed a Second Meal Effect by the Barley Leaves

During the snack, there was a moderate positive correlation in glucose AUC between the vehicle and barley groups (Figure 2A), a weak but not significant positive correlation between the vehicle and mulberry groups (Figure 2B) and a moderate positive correlation between the mulberry and barley groups (Figure 2C). These results suggest that barley leaves are equally effective in reducing glucose AUC in participants with a high or low glucose AUC following consumption of the biscuits, and that mulberry leaves are only able to decrease glucose AUC in participants with a high glucose AUC following consumption of the biscuits. 

During dinner, there was a moderate positive correlation in glucose AUC between the none and vehicle groups (Figure 2D), a moderate positive correlation between the none and mulberry groups, a weak but not significant positive correlation between the none and barley groups (Figure 2E,F), and a weak correlation between the mulberry and barley groups (Figure 2G). These results suggest that barley leaves are effective at lowering glucose AUC in participants with high glucose levels at dinner.

### 3.3. There Was No Relationship between Glucose AUCs at Snack and Dinner Times

There was no correlation of glucose AUC between the snack and dinner times among the none and vehicle groups (Figure 3A,B). In addition, there was no correlation of glucose AUC between the snack and dinner times for the mulberry and barley groups (Figure 3C,D). These results indicate that lowering glucose levels by a healthy snack does not impact glucose levels at dinner time, thereby negating the second meal effect.

### 3.4. Young Adults with Healthy Body Weights and Preferences for Evenings Had Higher Glucose Levels at Dinner Time

There were no significant correlations of glucose AUC among the various physiological factors at snack time (Table 5). However, the average glucose level on a daily basis was positively correlated with the glucose AUC for the none and mulberry groups at dinner time. In addition, BMI in the control group, as well as MEQ scores and MVPA in the vehicle and mulberry groups, were significantly correlated with glucose AUC at dinner. Dietary energy and dietary fiber were non-significantly positively correlated with glucose AUC in none and vehicle at dinner time. These results show that young adults with low BMI, high daily glucose levels and preferences for evenings have high glucose levels at dinner time.

### 3.5. A Second Meal Effect Was Observed at Dinner Time for Individuals with High Glucose Levels

There were individual differences in glucose AUC following the same dinner (Figure 4A), so participants were divided into high-(*N* = 9) and low-(*N* = 9) glucose groups. A clear second meal effect was observed at dinner time for the high-glucose group, but not for the low-glucose group (Figure 4B). These results demonstrate that by dinner time, individuals with high glucose levels experienced a strong reduction in glucose AUC if they had consumed the biscuit, or the biscuit plus mulberry or barley leaves as a snack beforehand. Additionally, increases in glucose level due to the biscuit were attenuated by simultaneous intake of mulberry or barley leaves for both high- and low-glucose groups.

### 3.6. Differences in Physiological Factors between the High- and Low-Glucose Groups

Given that a second meal effect was only observed for the high-glucose group, we decided to examine the physiological factors between the two groups to see if there were any differences (Figure 5A–H). The high-glucose group had a lower BMI and fat mass than the low-glucose group (Figure 5A,C), suggesting that lean participants have high glucose levels at dinner time. In addition, the high-glucose group appeared to consume more dietary energy and dietary fiber (Figure 5F,G).

## 4. Discussion

In the present study, we demonstrated that intake of powdered mulberry or barley leaves between lunch and dinner results in a second meal effect on glucose levels at dinner. Moreover, the simultaneous intake of mulberry or barley leaves significantly attenuated the biscuit-induced increase in glucose level during snack time. This finding aligns with that of previous reports [12,24,25], and further contributes to claims of the efficacy of dietary fibers and alpha-glucosidase in inhibiting increases in blood sugar incurred by high-glycemic foods [15,26,27].

Here, biscuit intake during the mid-afternoon blunted postprandial rises in glucose at dinner for participants with high glucose levels at dinner time. This result is very similar to that of earlier research [6,28], showing that eating a sweet snack between lunch and dinner ameliorates postprandial glycemic rises, compared with eating such snacks after dinner or lunch. In the current experiments, consumption of mulberry and barley leaves alongside biscuits at snack time resulted in a stronger second meal effect on glucose levels at dinner, compared to a biscuit alone.

Previous studies have reported that consumption of high dietary fiber and low glycemic-index meals provides a clear second meal effect [3,29,30]. Insoluble fiber influences postprandial glucose levels by accelerating the secretion of glucose-dependent insulinotropic polypeptide (GIP) [31], an incretin that stimulates postprandial insulin secretion. Not surprisingly, earlier research has reported decreases in postprandial glucose levels on account of cellulose intake [32,33]. Interestingly, there was no direct effect of glucose levels at snack time on those at dinner time. Indeed, the mechanism underlying the second meal effect has not been understood well up to this point. Following longer periods of fasting, it is known that free fatty acids (FFA) will increase [34], and higher FFA concentrations negatively affect insulin sensitivity [35]. Moreover, rodent experiments have demonstrated that mulberry and barley leaves modulate the diversity of microbiota and the production of short chain fatty acids [36,37], which are reported to be involved in insulin secretion and insulin sensitivity [38,39]. For our participants, it is possible that their increased glucose levels prevented the breakdown of fat and release of FFAs into the bloodstream. Other mechanisms, such as the secretion of insulin, glucagon and incretin [40], may be involved as well, warranting further investigation. Findings from our study showed that there are no direct similar mechanisms controlling blood glucose levels on snacking time and second meal time.

Both the mulberry and barley leaves had a similar impact on the second meal effect, though barley leaves were admittedly more potent. This data suggests that barley leaves are more effective at reducing postprandial glucose increases in people displaying a high glucose level at dinner time. Upon dividing the participants based on postprandial high- and low-glucose levels at dinner time, those with high glucose levels exhibited a very clear second meal effect following consumption of the biscuit, or biscuit with mulberry or barley leaves. This is intriguing, and implies that mulberry and barley leaves may be a good snack candidate for individuals with type 2 diabetes.

We were surprised to discover that glucose levels at dinner time were negatively correlated with BMI. The high-glucose group had a low BMI and low muscle volume, demonstrating that lean participants experience a more robust second meal effect. Furthermore, glucose levels at dinner were negatively correlated with a participant’s MEQ score, indicating that people who prefer evenings had higher levels of glucose following intake of no biscuits, a biscuit, or a biscuit with mulberry leaves. It is already known that lean people with little muscle often have hyperglycemia because they have less glucose uptake in their muscles [41,42], and those who prefer evenings display higher glucose levels at dinner time [43,44]. In the current experiment, we gave the same test meal at dinner, and therefore it is possible that the volume of dinner was too much for participants with low BMI. Thus, it is suggested that we should consider the advantage of the second meal effect when we have a chance to eat too much at dinner time.

The high-glucose group had high daily glucose levels and high diet energy, demonstrating that unhealthy people with high calorie meals and high blood glucose experience a more robust second meal effect. When we estimated dietary fiber intakes according to the FFQs, the high-glucose group was found to eat significantly higher amounts of dietary fiber compared with the low-glucose group. Habitual intake of dietary fiber may augment the response of the second meal effect to fiber and polysaccharides, such as those in the mulberry and barley leaves. Taken together with aforementioned discoveries, it is presumable that young adults with unhealthy feeding habits and preferences for evenings should be aware of the risk they face of postprandial increases in glucose at dinner time, but if they eat a healthy snack beforehand, we expect it will exert a strong second meal effect on glucose control.

The present study had limitations and strengths. First, our participants were predominantly all healthy. Thus, our findings do not apply to other demographics, such as elderly people and patients with type 2 diabetes. Second, the sample size in this study was too small to draw conclusions about the effects of the snack on postprandial glucose levels. Future research should aim to investigate this effect with a larger number of test participants. Third, breakfast and lunch were not controlled for during the study period. Therefore, the effects of lunch intake on change of postprandial blood glucose levels cannot be denied, though participants were instructed to maintain their normal lifestyles during the study. Fourth, we did not control the dinner volume individually. Fifth, it will be necessary to examine the effect of snacks on postprandial blood glucose levels in daily life where dinner is not controlled. Sixth, we did not examine the impact of mulberry or barley leaves, alone, on glucose levels, so can only indirectly infer how they attenuated rises in glucose when consumed simultaneously with biscuits. As a strength, the present findings are very novel, and open a new avenue for controlling glucose levels.

## 5. Conclusions

Our findings reveal that eating snacks alongside mulberry or barley leaves is an effective way to suppress postprandial high glucose levels in young adults with high glucose levels and lean body weights, and who prefer evenings. Future investigations should apply these results to middle aged persons using different types of supplements.

## Figures and Tables

**Figure 1 nutrients-12-01580-f001:**
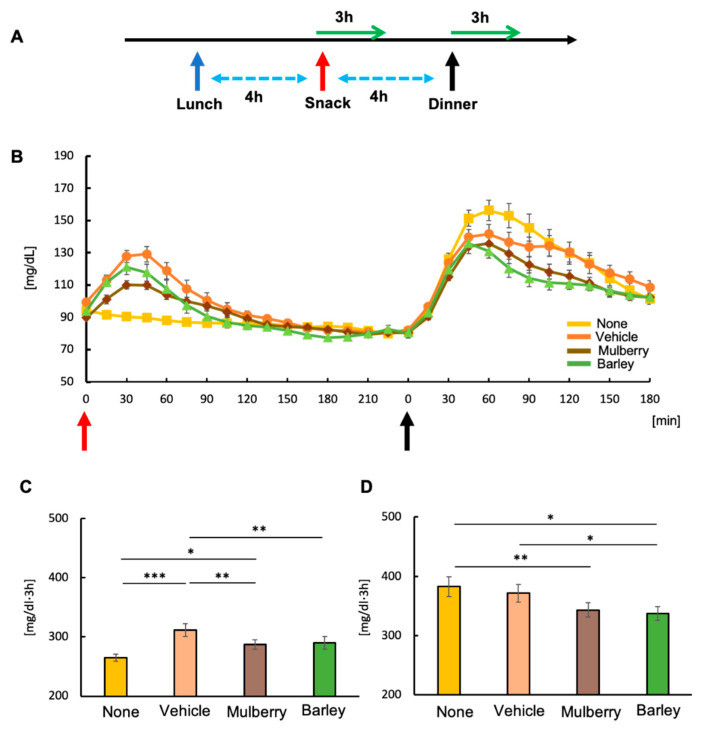
Changes in blood glucose levels. Study protocol (**A**), changes in glucose levels after the snack and dinner (**B**), area under the curve (AUC) of glucose levels 180 min after snack intake (**C**) and dinner intake (**D**). Values are means ± standard errors. * *p* < 0.05, ** *p* < 0.01, *** *p* < 0.001.

**Figure 2 nutrients-12-01580-f002:**
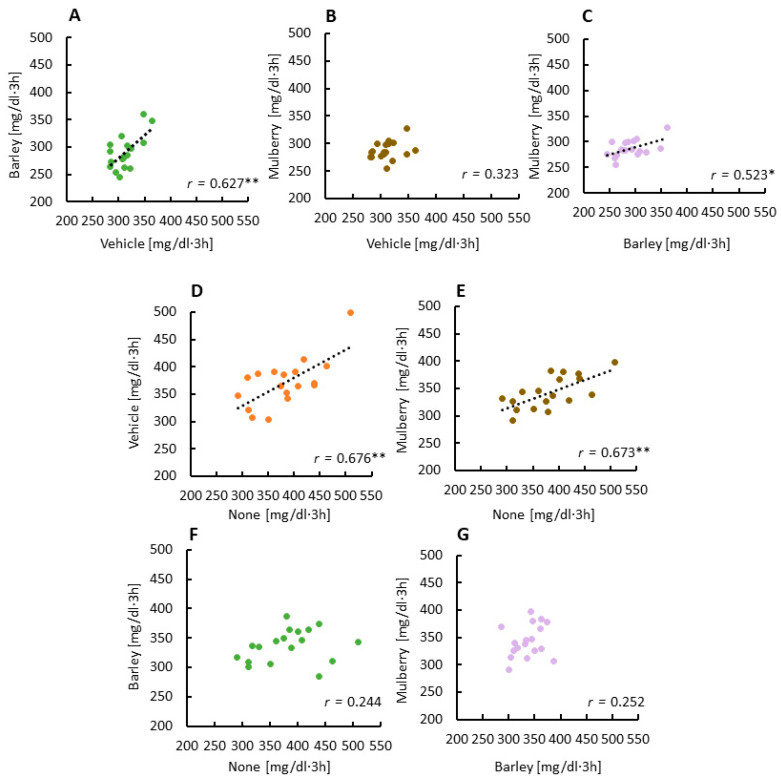
The relationship between area under the curve (AUC) glucose levels between each session. Correlations of glucose AUC between the vehicle and barley groups at snack time (**A**), correlations of glucose AUCs between the vehicle and mulberry groups at snack time (**B**), correlations of glucose AUC between the barley and mulberry groups at snack time (**C**), correlations of glucose AUC between the none and vehicle groups at dinner time (**D**), correlations of glucose AUC between none and mulberry groups at dinner time (**E**), correlations of glucose AUC between the none and barley groups at dinner time (**F**), correlations of glucose AUC between the barley and mulberry groups at dinner time (**G**). * *p* < 0.05, ** *p* = 0.01 (**A**,**D**,**E**).

**Figure 3 nutrients-12-01580-f003:**
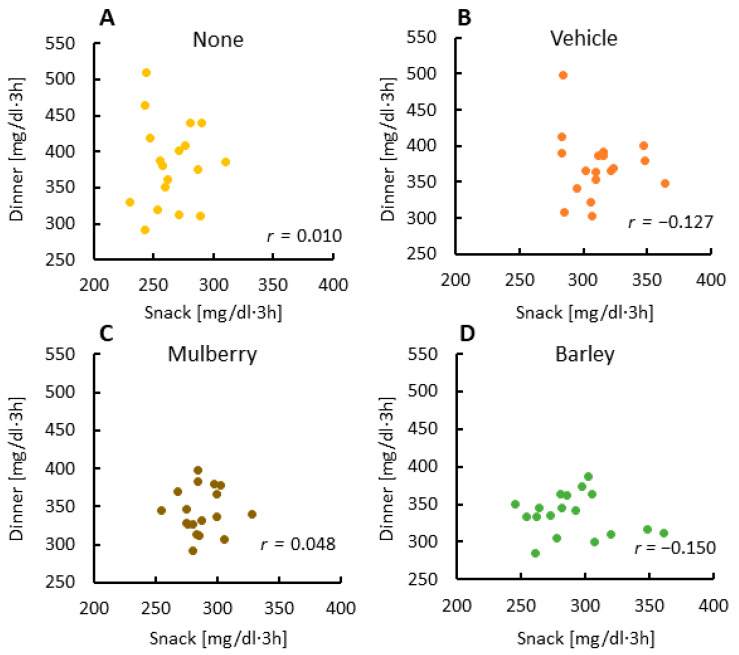
The relationship between area under the curve (AUC) glucose levels at the snack and dinner times for each session. Correlations of glucose AUC between the snack and dinner times for the none group (**A**), correlations of glucose AUC between the snack and dinner times for the vehicle group (**B**), correlations of glucose AUC between the snack and dinner times for the mulberry group (**C**), correlations of glucose AUC between the snack and dinner times for the barley group (**D**).

**Figure 4 nutrients-12-01580-f004:**
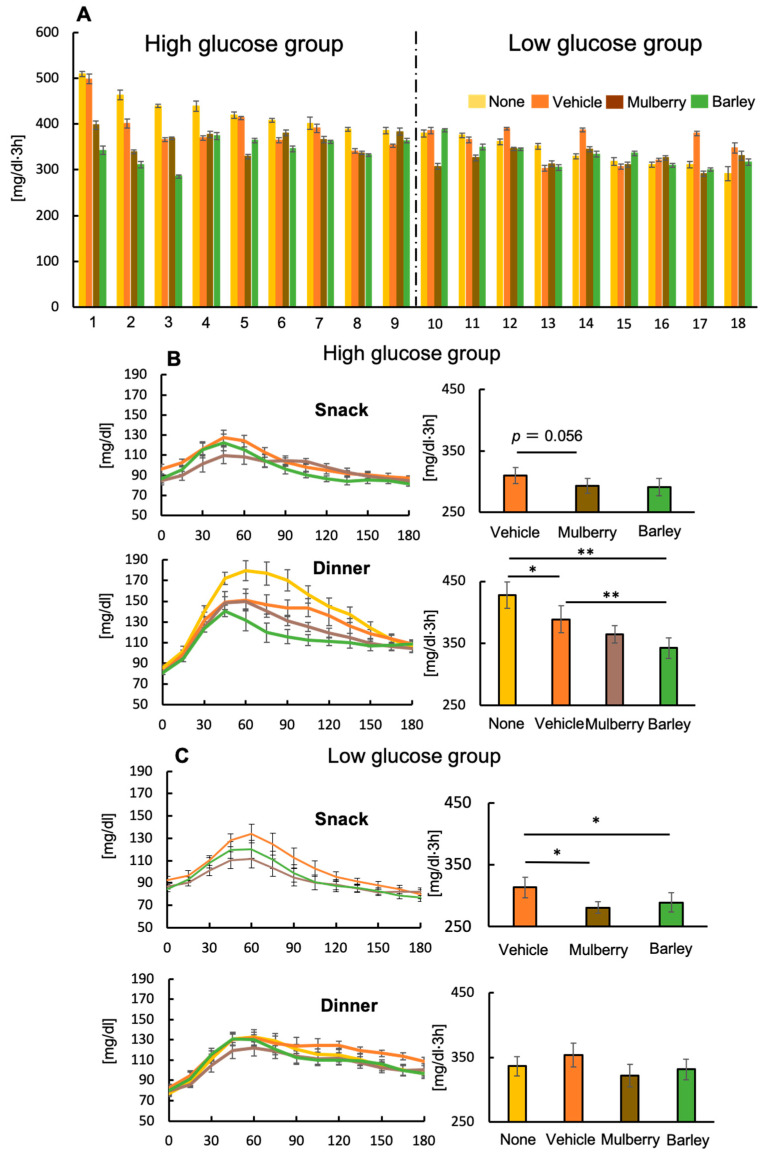
Changes in glucose levels for the high- and low-glucose groups. Area under the curve (AUC) of glucose levels after dinner for the none group (**A**), changes in glucose levels and AUC after consuming the snack and dinner for the high-glucose group (**B**), changes in glucose levels and AUC after consuming the snack and dinner for the low-glucose group (**C**). Values are means and standard errors. * *p* < 0.05, ** *p* < 0.01.

**Figure 5 nutrients-12-01580-f005:**
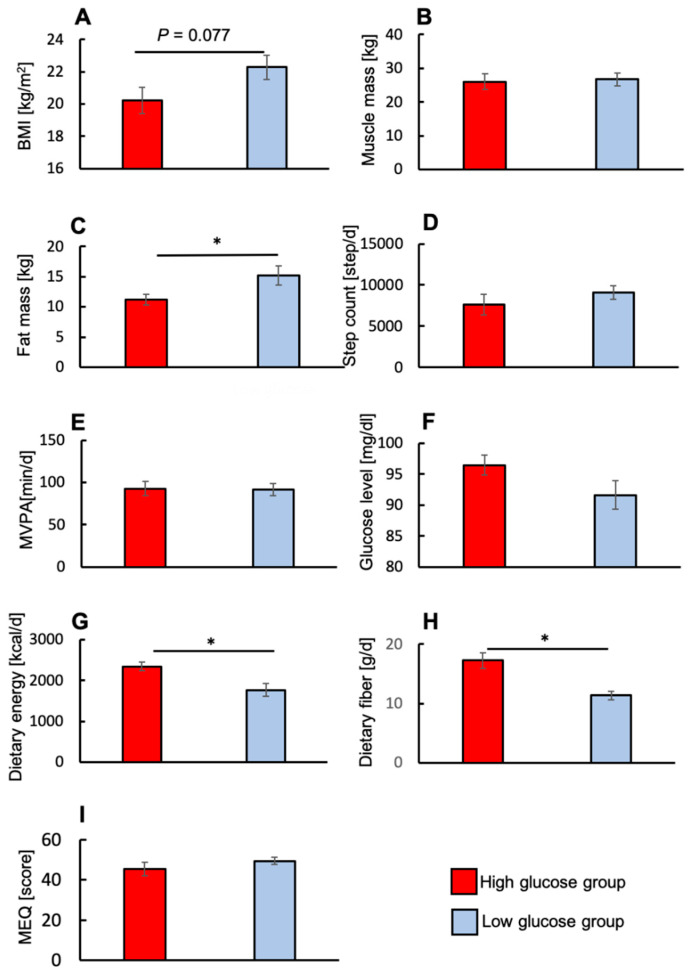
Differences in physiological factors for the high- and low- glucose groups. Body Mass Index (BMI) (**A**, *N* = 18), muscle mass (**B**, *N* = 18), fat mass (**C**, *N* = 18), step count (**D**, *N* = 18), moderate-to-vigorous physical activity (MVPA) (**E**, *N* = 18), glucose level on daily basis (**F**, *N* = 15), dietary energy (**G**, *N* = 15), dietary fiber (**H**, *N* = 15), Morningness–Eveningness Questionnaire (MEQ) (**I**, *N* = 18). Values are means ± standard errors. * *p* < 0.05.

**Table 1 nutrients-12-01580-t001:** Meal time in each session.

Meal Time	None	Vehicle	Mulberry	Barley
Lunch	12:48 ± 0:10	12:53 ± 0:08	12:53 ± 0:11	12:59 ± 0:10
Snack	-	16:56 ± 0:08	16:56 ± 0:11	17:03 ± 0:09
Dinner	20:54 ± 0:11	20:59 ± 0:09	20:58 ± 0:10	21:06 ± 0:09

**Table 2 nutrients-12-01580-t002:** Nutrition facts for powdered mulberry and barley leaves (for a 3 g serving).

Nutrients	Mulberry	Barley
Energy (kcal)	11.3	10.7
Protein (g)	0.7	0.8
Fat (g)	0.2	0.2
Total Carbohydrate (g)	1.7	1.5
Sugar (g)	0.6	0.3
Dietary fiber (g)	1.1	1.2

**Table 3 nutrients-12-01580-t003:** Nutrition facts for the biscuit and dinner.

Nutrients	Biscuit	Dinner
Energy (kcal)	156.6	752.0
Protein (g)	3.0	22.3
Fat (g)	4.7	33.6
Total Carbohydrate (g)	25.9	91.4
Sugar (g)	25.2	87.3
Dietary fiber (g)	0.7	4.1

**Table 4 nutrients-12-01580-t004:** Study participants characteristics (*N* = 18).

	Baseline	After Two Weeks
Age (y)	23.1 ± 0.4	
Height (cm)	166.1 ± 2.1	
Weight (kg)	59.3 ± 2.5	59.0 ± 2.4
BMI (kg/m^2^)	21.2 ± 0.6	21.1 ± 0.5
Fat (%)	21.8 ± 1.5	20.8 ± 1.5
Muscle mass (kg)	25.8 ± 1.4	25.7 ± 1.4
Fat mass (kg)	12.8 ± 0.9	13.2 ± 1.0
Step count (steps/d)	9100.6 ± 817.8	8973.3 ± 673.4
MVPA (min/d)	89.5 ± 7.8	90.3 ± 6.2
MEQ	47.8 ±1.8	
Energy intakes (kcal/d) ^(1)^	2036.4 ± 122.6	
Dietary fiber intakes (g/d) ^(1)^	14.0 ± 1.1	

All data are presented as mean ± standard error. BMI: body mass index, MVPA: moderate-to-vigorous physical activity, MEQ: Morningness–Eveningness Questionnaire. ^(1)^
*N* = 15.

**Table 5 nutrients-12-01580-t005:** Correlation between physiological factors and snack/dinner glucose levels.

	Snack			Dinner			
	Vehicle	Mulberry	Barley	None	Vehicle	Mulberry	Barley
**Glucose level (mg/dL)**	−0.34	0.07	−0.37	0.54 *	0.24	0.49 *	0.25
**BMI (kg/m^2^)**	0.00	−0.13	−0.07	−0.56 *	−0.43 ^(1)^	−0.40	−0.10
**Muscle mass (kg)**	−0.07	−0.02	−0.24	−0.24	−0.44 ^(2)^	0.45 ^(3)^	0.28
**Fat mass(kg)**	−0.15	−0.19	−0.2	−0.43	−0.23 ^(4)^	−0.45 ^(5)^	0.00
**Step count (steps/d)**	0.08	−0.20	0.26	−0.11	0.36	−0.20	0.24
**MVPA (min/d)**	−0.33	−0.09	0.37	0.02	0.52 *	0.05	0.41
**MEQ (score)**	0.14	0.09	0.21	−0.48 ^(6)^	−0.60 *	−0.56 *	−0.24
**Dietary energy(kcal/d)**	0.18	0.20	0.38	0.32	0.49	0.02	0.32
**Dietary fiber (g/d)**	−0.13	−0.07	0.06	0.42	0.39	0.09	0.36

Glucose level was averaged on daily basis. * *p* < 0.05 (Pearson’s correlation coefficient). ^(1)^
*p* = 0.073, ^(2)^
*p* = 0.066, ^(3)^
*p* = 0.059, ^(4)^
*p* = 0.076, ^(5)^
*p* = 0.063, ^(6)^
*p* = 0.052.

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
