# Peer review of "Consumption of Biscuits with a Beverage of Mulberry or Barley Leaves in the Afternoon Prevents Dinner-Induced High, but Not Low, Increases in Blood Glucose among Young Adults"

_nutrients, 2020, doi:10.3390/nu12061580_

Round 1

Reviewer 1 Report

Thank you very much for interesting kind of work. You structured it very well.

Here are my questions and comments:

INTRODUCTION

  • You described a biscuit as a “healthy” snack. Can you explain, why it is healthy? Wouldn`t it be possible to use fruits as a healthy snack?

2.3. TEST SESSION

  • How long is a test session? How many days did the participants stay on one of the four regimes?
  • Please explain the time during the day for lunch, snack, dinner and evening. There Might be big cultural differences of your readers.
  • What do you mean by “4 h after or 4 h before dinner”? According to your figure 1A it should be “4 h after and 4 h before dinner” (line 119). Or did I misunderstand your study design?

2.5. DIETARY INTAKE ASSESSMENT

  • Why do you use a Food Frequency Questionnaire since the participants get a standardized test dinner (line 101) during the evaluation?

3.2. CORRELATIONS….

  • line 178 you are talking about participants with high glucose levels. At this point you did not introduce this group of participants.

3.4. YOUNG ADULTS ….

  • What do you mean by “evening”? Are you talking about late “evening meals”? Is that later than dinner time? This question might be already answered when you work on 2.3.

3.5. A SECOND MEAL ….

  • If you divide your participants in two groups, you have to explain it in the method section.
  • Do the two groups present significant differences in blood glucose-levels? If so, please show the data. Since you have only a small number of patients the differences in your findings could be accidentally and not in relation to the blood glucose.

3.6. DIFFERENCES IN …..

  • Why did the participants have a different nutrient intake while they are getting standardized meals? You might have answered this question working on 2.5.

Reviewer 2 Report

Dear editors,

The proposed work is novel, addresses a topic of great interest in the field of nutrition, and is interesting for the journal's audience.

Although the sample is somewhat small, it is interesting as a pilot study, and the authors acknowledge the study's main limitations. The methodological approach and statistical analysis seem adequate and, in general, the results are well described and discussed.

I thank you for your confidence in me as a reviewer, 

Best regards.
